# Reliability and O&M key performance indicators of onshore and offshore wind turbines based on field-data analysis

Julia Walgern<sup>1,2</sup>, Nils Stratmann<sup>1,3</sup>, Martin Horn<sup>1,3</sup>, Nathalene WY Then<sup>1,3</sup>, Moritz Menzel<sup>1,3</sup>, Fraser Anderson<sup>1,4</sup>, Athanasios Kolios<sup>2,5</sup>, Katharina Fischer<sup>1</sup>

- 5 <sup>1</sup>Fraunhofer Institute for Wind Energy Systems IWES, 30159 Hannover, Germany
  - <sup>2</sup>University of Strathclyde, 16 Richmond St, Glasgow G1 1XQ, United Kingdom
  - <sup>3</sup>Leibniz University Hannover, Welfengarten 1, 30167 Hannover. Germany
  - <sup>4</sup>Fraunhofer UK Research Ltd., 99 George St., Glasgow G1 1RD, United Kingdom
  - <sup>5</sup>Technical University of Denmark, Department of Wind & Energy Systems, Risø Campus, Frederiksborgvej 399, 4000 Roskilde, Denmark

Correspondence to: Julia Walgern (julia.walgern@iwes.fraunhofer.de)

Abstract. Based on maintenance data from over 1000 onshore and offshore wind turbines covering more than 4200 operating years, this study presents an analysis of failure rates, repair times, and maintenance resource requirements, focusing on subsystem-level reliability. Failure rates per turbine and megawatt are compared and failure behaviour over time is examined. Next to failure events, further corrective and preventive maintenance interventions are analysed. To provide more detailed insights for operation and maintenance simulations, a distinction is made between total major component replacements and those specifically requiring a jack-up vessel. Results show that onshore wind turbines have higher failure rates per megawatt than offshore wind turbines. Key subsystems including the pitch system, the control system, and power converter system are identified as critical to overall wind turbine reliability for both onshore and offshore wind turbines. For the overall wind turbine system, a failure behaviour over time following a bathtub curve is identified, with distinct trends for individual subsystems.

#### 1 Introduction

The global shift towards renewable energy has driven significant investments in wind energy, positioning it as a cornerstone of sustainable power generation. With the growing reliance on wind energy, especially in offshore environments, the reliability and performance of wind turbines have become critical factors that directly influence energy yield, operational costs, and overall asset integrity (Ioannou et al., 2018). The effective management of these assets is particularly crucial as the industry aims to optimise operational efficiency and minimise downtime. However, achieving this requires a profound understanding of failure mechanisms and maintenance needs, underpinned by reliable data (Martinez-Luengo et al., 2019; Wang et al., 2023; Khan and Byun, 2024).

While there has been considerable progress in the development of wind turbine (WT) technology, the reliability assessment and optimisation of operations and maintenance (O&M) of these systems has often been hampered by a lack of comprehensive, high-quality field data. Existing studies on wind turbine reliability mostly rely on limited datasets or combine data from diverse

turbine types and operating conditions without sufficient granularity (Carroll et al., 2016; Li and Guedes Soares, 2022). Such approaches can obscure the differences in reliability performance across turbine types, manufacturers, and environmental contexts. Consequently, there is a critical need for detailed analyses based on comprehensive field data that can provide more accurate and actionable insights into failure rates and maintenance strategies (Cevasco et al., 2021).

This study represents a significant advancement in the field by presenting an extensive analysis of wind turbine reliability based on a large, representative sample of field data. Drawing on maintenance reports spanning more than 4,200 operational years from both onshore and offshore wind turbines, this research provides one of the most comprehensive evaluations of failure rates and further O&M-related key performance indicators (KPIs) to date. The analysis includes data from nine different onshore and four offshore wind turbine original equipment manufacturers (OEMs), covering a range of turbine capacities and operational contexts. This breadth and depth of data allow for a more comprehensive understanding of reliability performance across various wind turbine systems, informing both design optimisation and O&M strategies for future wind farms.

Furthermore, the study introduces a detailed categorisation of wind turbine failures using the reference designation system RDS-PP, which is applied systematically for standardised component classification across all different turbine types and designs. By focusing on system and subsystem level and calculating average failure rates along with corresponding confidence intervals, this work studies the reliability behaviour of wind turbines with a rare and considerable level of detail, providing unnormalised KPIs and uncertainty quantifications. The findings reveal detailed insights into the different failure rates of onshore versus offshore turbines, the impact of turbine rated power on reliability, and the temporal patterns that characterise wind turbine failures.

This level of granularity in data analysis not only enhances the reliability modelling of current wind turbine fleets but also serves as a valuable resource for OEMs, operators, and policymakers looking to improve the design and operation of future wind farms. The analysis conducted in this study highlights specific reliability challenges as well as opportunities for technological improvements and maintenance optimisation, making it an essential input for risk management and decision-making in the wind energy sector (Ayyildiz and Erdogan, 2024).

Recognising the sensitivity of the data involved, we have systematically evaluated and implemented measures to ensure the confidentiality and security of the datasets used in this research. These measures were crucial for protecting proprietary information and maintaining the trust of data providers while enabling the comprehensive analysis presented herein.

In the sections that follow, we provide a thorough review of the current state of the art in wind turbine reliability research and outline the methodologies and datasets used in this study. This is followed by an in-depth presentation of the results, discussing

their implications for both the operational management of wind farms and future research directions.

## 2 State of the art literature on wind turbine reliability

# 2.1 Overview of wind turbine reliability research

Understanding wind turbine reliability is crucial for optimising their performance and minimising operational costs, especially for offshore installations. Early studies primarily focused on analysing key reliability metrics such as average failure rates, mean time to repair, and availability, which are essential for developing maintenance strategies (Tavner et al., 2012). Over the years, efforts have been made to standardise the collection and analysis of reliability, availability, and maintainability (RAM) data across diverse turbine types and environments (Cevasco et al., 2021). Prominent initiatives like the WInD-Pool common knowledge base in Germany and the SPARTA program in the United Kingdom have been instrumental in adopting structured methodologies to gather operational data from wind farms (Leahy et al., 2019; Fraunhofer IEE, 2018).

Due to the strict confidentiality of maintenance data, only a limited number of reliability studies have been published. European initiatives such as WMEP (Hahn et al., 2007), as well as WSD, WSDK, and LWK (e.g. (Tavner et al., 2007; Spinato et al., 2009)) were among the first to analyse WT maintenance data, covering periods from the 1990s until 2004. More recent studies include those from the Reliawind project (Gayo, 2011), the University of Strathclyde (Carroll et al., 2016), the AWESOME project (Reder et al., 2016), and the SPARTA initiative (SPARTA, 2017). Additionally, detailed reviews of published failure rate statistics have been conducted by (Pfaffel et al., 2017; Cevasco et al., 2021; Artigao et al., 2021; Turnbull et al., 2022). However, as most of these studies rely on data sets recorded before 2015, with SPARTA being the only initiative providing more recent reliability and performance KPIs from 2020/21 (SPARTA, 2022), there remains a need for comprehensive and high-quality field data, particularly for modern, larger turbines.

# 2.2 Key performance indicators for reliability and O&M assessment

KPIs are critical for evaluating wind turbine reliability and optimising O&M. Among the most widely used KPIs in reliability studies are failure rates, mean time to failure (MTTF), mean time to repair (MTTR), and time-based or energy-based availability (Pfaffel et al., 2017). An overview of commonly used KPIs is presented in Table 1. The failure rate, typically measured as the number of failures per turbine per year, is a fundamental metric that provides important insights into turbine reliability and is typically utilised as input for O&M modelling (Donnelly et al., 2024).

Comparative studies reveal significant differences not only in the aforementioned failure rates but also in further O&M-related KPIs between onshore and offshore wind turbines and associate these with varying environmental conditions, maintenance access, and design complexities (Faulstich et al., 2011). Subsystems such as the pitch system, hydraulic systems, rotor, power converter system, generator, and gearbox are often identified as having the highest failure rates (cf. (Carroll et al., 2016; SPARTA, 2017)), especially in offshore installations where repairs are more challenging. (Scheu et al., 2017) highlighted that corrective maintenance for these critical components often results in substantial downtime, underscoring the need for robust designs and advanced monitoring systems.

Identifying general trends in reliability and maintainability helps operators to pinpoint where reliability improvements could lower the levelized cost of energy (LCOE). However, challenges such as inconsistent data collection practices complicate the comparison of reliability metrics across different studies. Efforts like those by the International Energy Agency (IEA) Wind Task 33 aim to address these challenges by providing standardised frameworks for data collection and analysis (Hahn et al., 2017).

The variability in methodologies and reliability indicators points to the need for more standardised approaches to provide actionable insights. Enhancing RAM databases with detailed failure and operational data is crucial for advancing wind turbine design and maintenance strategies (Leimeister and Kolios, 2018).

Table 1: Summary table of key performance indicators (KPIs) for wind turbine reliability.

| KPI                                | Definition                                                                                             | Importance                                                                                                                                                                      | <b>Common Calculation Methods</b>                |
|------------------------------------|--------------------------------------------------------------------------------------------------------|---------------------------------------------------------------------------------------------------------------------------------------------------------------------------------|--------------------------------------------------|
| Failure Rate                       | Average number of failures per unit (e.g., per turbine) per year for a specific component or subsystem | Indicates the reliability of wind turbine components and subsystems; high failure rates can lead to increased maintenance costs and downtime                                    | Empirical analysis using maintenance data        |
| Corrective<br>Maintenance<br>Rate  | Frequency of corrective maintenance interventions                                                      | Indicates the reliability of wind turbine components and subsystems                                                                                                             | Empirical analysis of maintenance records        |
| Unscheduled<br>Maintenance<br>Rate | Frequency of maintenance activities related to unexpected failures                                     | High rates suggest frequent unexpected failures and may affect downtime and operational planning                                                                                | Empirical data analysis from maintenance reports |
| Preventive<br>Maintenance<br>Rate  | Frequency of preventive maintenance interventions                                                      | Helps in understanding the maintenance<br>strategy; high rates suggest that<br>preventive maintenance interventions and<br>associated costs are accepted to prevent<br>failures | Empirical data analysis from maintenance reports |
| Mean Time<br>to Failure<br>(MTTF)  | Average time to failure for a non-repairable specific component or subsystem                           | Helps in understanding the expected<br>lifespan of components; a higher MTTF<br>indicates better reliability                                                                    | Empirical data analysis from maintenance reports |

110

115

| Mean Time                    | Average time required to                                                                                   | Critical for planning maintenance                                                                                                                                                        |                                                                         |
|------------------------------|------------------------------------------------------------------------------------------------------------|------------------------------------------------------------------------------------------------------------------------------------------------------------------------------------------|-------------------------------------------------------------------------|
| to Repair                    | repair a failed component or                                                                               | resources and minimising downtime; a                                                                                                                                                     | Empirical analysis based on                                             |
| (MTTR)                       | subsystem and restore it to                                                                                | lower MTTR indicates more efficient                                                                                                                                                      | maintenance records                                                     |
| (MITIK)                      | operational condition                                                                                      | maintenance processes                                                                                                                                                                    |                                                                         |
| Mean Time                    | Average time between                                                                                       | In disease the malichility of sain describes                                                                                                                                             |                                                                         |
| Between                      | successive failures of a                                                                                   | Indicates the reliability of wind turbine                                                                                                                                                | Calculated as the inverse of the                                        |
| Failures                     | repairable system or                                                                                       | components and subsystems; a higher                                                                                                                                                      | average failure rate                                                    |
| (MTBF)                       | component                                                                                                  | MTBF indicates better reliability                                                                                                                                                        |                                                                         |
|                              |                                                                                                            |                                                                                                                                                                                          |                                                                         |
|                              | The proportion of time a                                                                                   | Reflects overall performance and                                                                                                                                                         | Time-based calculations using                                           |
| Availability                 | The proportion of time a wind turbine is operational                                                       | Reflects overall performance and reliability of wind turbines; high                                                                                                                      | Time-based calculations using operational and downtime data,            |
| Availability<br>(time-based) |                                                                                                            | •                                                                                                                                                                                        |                                                                         |
| ·                            | wind turbine is operational                                                                                | reliability of wind turbines; high                                                                                                                                                       | operational and downtime data,                                          |
| ·                            | wind turbine is operational and capable of generating                                                      | reliability of wind turbines; high<br>availability is key to maximising energy<br>production and minimising losses                                                                       | operational and downtime data, i.e. typically SCADA data;               |
| (time-based)                 | wind turbine is operational<br>and capable of generating<br>power                                          | reliability of wind turbines; high availability is key to maximising energy production and minimising losses  Directly impacts energy yield and                                          | operational and downtime data, i.e. typically SCADA data; Markov models |
| ·                            | wind turbine is operational<br>and capable of generating<br>power<br>Total time during which a             | reliability of wind turbines; high availability is key to maximising energy production and minimising losses  Directly impacts energy yield and economic returns; high downtime leads to | operational and downtime data, i.e. typically SCADA data; Markov models |
| (time-based)                 | wind turbine is operational and capable of generating power  Total time during which a wind turbine is not | reliability of wind turbines; high availability is key to maximising energy production and minimising losses  Directly impacts energy yield and                                          | operational and downtime data, i.e. typically SCADA data; Markov models |

#### 2.3 Common causes of failures and reliability challenges reported in literature

Understanding the prevailing causes of failures in wind turbines is crucial for enhancing their reliability and maintenance strategies. In the literature, the following failure modes and causes are reported: The gearbox frequently fails due to bearing and gear fatigue, misalignment, and lubrication issues, leading to significant downtime (Carroll et al., 2016; Reder et al., 2016). Additionally, tribological failures such as pitting and scuffing affect gearboxes due to inadequate lubrication. The generator faces electrical and mechanical failures such as stator faults and insulation degradation due to electrical surges and thermal stresses (Kavakli and Gudmestad, 2023). Power converter failures are dominated by failures of the power semiconductor modules, their driver boards, the converter control system as well as the cooling system (Fischer et al., 2019a; Fischer et al., 2023). The pitch system is vulnerable to mechanical wear from continuous blade angle adjustments in varying wind conditions (Li et al., 2022). Meanwhile, blades are prone to erosion, fatigue, and lightning strikes, affecting turbine performance (Lopez and Kolios, 2022).

Common failure mechanisms include fatigue, particularly in moving parts like blades and gear teeth due to cyclic loading. For many years, fatigue due to power and thermal cycling was postulated to be the main failure mechanism also in power converters, until comprehensive field-data and damage analyses revealed that climatic influences, which drive corrosion and affect insulation integrity in the converter, play a more important role in the wind-power application (Fischer et al., 2019a;

https://doi.org/10.5194/wes-2025-212 Preprint. Discussion started: 24 October 2025

© Author(s) 2025. CC BY 4.0 License.

135

140

Fischer et al., 2019b). Corrosion is a relevant failure mechanism also for support structures, especially in offshore environments where saltwater accelerates degradation (Scheu et al., 2019; Tremps et al., 2024).

It is important to keep in mind that the detailed identification of failure root causes and the underlying mechanisms can be a complex and laborious task, often requiring comprehensive data evaluation and analyses of damaged components. As the above example of power converters shows, there is a certain risk that hypotheses or postulates about prevailing failure mechanisms propagate through the literature and divert attention from the reality observed in the field.

## 2.4 Impact of turbine design, manufacturer and age on reliability

The reliability of wind turbines is significantly influenced by their design and the manufacturer. Studies have shown that design choices, such as drivetrain configurations (e.g., geared vs. direct drive) and control systems, affect failure rates and maintenance needs (Carroll et al., 2015; Carroll et al., 2018). For instance, direct-drive turbines eliminate the gearbox, reducing failures associated with gears and bearings, but they may have higher rates of electrical component failures due to the larger size and complexity of the generator and converter systems. Additionally, differences in manufacturing quality and component selection between manufacturers can lead to variability in reliability performance (Dahane et al., 2015). Standardisation, stringent quality control during the design and manufacturing phases as well as test-based reliability validation are essential to reduce such variability, ensuring consistent reliability across different turbine models and brands.

The operating age of wind turbines also significantly impacts their reliability. As turbines age, wear and tear from continuous operation, exposure to harsh environmental conditions, and fatigue loading can lead to increased failure rates (Tavner et al., 2012). Studies indicate that older turbines often experience failures in components such as blades, gearboxes, and electrical systems, which degrade due to prolonged exposure to mechanical stresses and environmental factors like temperature and humidity variations (Le and Andrews, 2015). Other subsystems, such as the power converter, exhibit pronounced early failures (Anderson et al., 2025). In general, failure patterns of technical systems typically follow a "bathtub curve", where failure rates are decreasing during the early-failure phase, remain relatively constant during a "useful life" phase, and increase again as components degrade in the deterioration phase (Rigdon and Basu, 2000). Understanding these patterns is crucial for optimising maintenance strategies and extending the operational life of wind turbines.

# 2.5 Data-driven approaches and advanced analytical methods

The use of big data and machine learning (ML) has transformed the field of wind turbine reliability analysis, enabling more accurate early fault detection and enhanced maintenance strategies. Recent advancements leverage data-driven approaches using large datasets from SCADA (Supervisory Control and Data Acquisition) systems, which provide high-frequency data on turbine operations and performance (Zaher et al., 2009; Encalada-Dávilla et al., 2021). Machine learning techniques such as neural networks, random forests, and support vector machines have been employed to detect patterns in operational data, predict failures, and optimise maintenance schedules, thereby reducing downtime and maintenance costs (Black et al., 2021; Kusiak and Verma, 2011; Lorenzo-Espejo et al., 2022). AI-based predictive maintenance approaches also incorporate data

165

170

fusion techniques that combine SCADA data with environmental and maintenance records, offering a more comprehensive view of turbine health and enabling proactive interventions (Jeong et al., 2020).

Recent meta-analyses and systematic reviews have consolidated findings across multiple studies to provide higher-level insights into wind turbine reliability management. For example, a meta-analysis by (Dao et al., 2019) aggregated reliability data from diverse sources, revealing trends in failure rates and highlighting critical components that require attention. These reviews often use statistical methods to compare data from different regions, turbine types, and operating conditions, offering a benchmark for reliability performance. By synthesising data from various studies, systematic reviews inform best practices for condition monitoring, component design, and maintenance planning, addressing gaps in existing literature and guiding future research. Such efforts help standardise reliability metrics and improve the robustness of reliability models, ensuring more effective asset management strategies for both onshore and offshore wind farms.

### 160 2.6 Knowledge gaps and contribution of the present study

Despite significant advancements in wind turbine reliability research, several gaps remain. A summary of those is shown in Table 2. Many studies rely on limited sample sizes and data from specific regions, which may not accurately represent broader operational contexts (Leahy et al., 2019). There is also a lack of comprehensive field data that captures the full spectrum of failure modes and environmental influences, especially for offshore turbines (Cevasco et al., 2021). This is often related to strict data confidentiality. Additionally, existing research often focuses only on a few subsystems (e.g. (SPARTA, 2017)), leading to gaps in reliability modelling. For example, (Hart et al., 2020) highlight that main bearings are frequently overlooked in reliability analyses. In reliability analyses, ensuring the recentness of data and coverage of modern WT technology remains a key challenge. As a result, many studies frequently reference literature based on older data sets that primarily reflect outdated turbine technology.

Table 2: Summary table of research gaps.

| Research Gap               | Description                                                                                        |
|----------------------------|----------------------------------------------------------------------------------------------------|
| Limited sample sizes       | Many studies use data from small, specific samples, limiting the generalisability of the findings. |
| Lack of diversity in field | Inadequate data coverage on different environments and conditions, especially for offshore sites   |
| data                       |                                                                                                    |
| Insufficient coverage of   | Underrepresentation of specific subsystem failure types or insufficient reliability data for       |
| certain subsystems         | certain subsystems                                                                                 |
| Lack of recent field data  | Most studies are based on old data sets, not covering modern WT technology.                        |
| Need for standardisation   | Lack of standard methodologies and definitions across studies complicates comparative              |
| and harmonisation          | analysis.                                                                                          |

https://doi.org/10.5194/wes-2025-212 Preprint. Discussion started: 24 October 2025

© Author(s) 2025. CC BY 4.0 License.

This study aims to address these gaps by using a more representative sample size and conducting a comprehensive analysis of both onshore and offshore wind turbine maintenance data. By integrating diverse datasets including modern turbine technology and systematically evaluating failure modes across various subsystems, this research offers a more holistic view of turbine reliability which is applicable for future wind farm design and operation.

#### 3 Methodology and evaluated data sets

# 3.1 Methodology

#### 3.1.1 Field-data collection and pre-processing

Maintenance reports, which are available for each visit of a wind turbine, of more than 1000 wind turbines were collected making an effort to include a variety of turbine types of both onshore and offshore turbines. Attention was paid to incorporate recently commissioned turbines as well as having data sets of turbines which have a certain track record already. This leads to a unique field-data collection with respect to its size, diversity and recentness.

Maintenance records include information about what maintenance intervention was carried out on which turbine on which date. Those reports can have different lengths and levels of detail. Typically, at least spare parts and / or work descriptions are recorded which allow one to understand what kind of work technicians have performed on the turbines. In order to conduct different reliability analyses, the data needs to be machine-readable and comparable even though the reports stem from different organisations and sites. Within this study standards and guidelines like the reference designation system RDS-PP for wind turbines (VGB PowerTech, 2014) and the State-Event-Cause-Code "ZEUS" (FGW e.V. - Fördergesellschaft Windenergie und andere Erneuerbare Energien, 2013) are utilised to support the pre-processing. RDS-PP is used to classify maintenance interventions according to the components and subsystems that were maintained. Using ZEUS, activities performed by technicians are labelled as corrective and preventive and further differentiated according to the specific maintenance action undertaken. The pre-processing results in a comprehensive field-data base covering:

• Wind turbine ID and respective wind farm

- Wind turbine manufacturer and type
- Commissioning date of the turbine
- Rated power of the turbine
- Technical information about the different subsystems
- Coordinates of the turbine
  - Data provider
  - Time stamps of start and end date of each maintenance activity
  - Number of technicians involved
  - Components and subsystems affected (standardised codes of RDS-PP)

• Type of maintenance activity (standardised codes of ZEUS)

# 3.1.2 Reliability analyses

In order to assess O&M activities and WT reliability performance, different reliability analyses are performed and KPIs computed. Respective KPIs can be utilised for benchmarking of different assets, understanding failure patterns as a basis for developing countermeasures, or as input for development and O&M simulation of future wind farms.

KPIs are assessed for corrective and preventive maintenance interventions. Particular attention is paid to failures of components and subsystems as those are afflicted with costly downtimes requiring maintenance and the use of spare parts. Within this study, a failure is defined as an event necessitating corrective maintenance (ZEUS code "02-08-01") and which is not resettable but requires a component to be replaced (ZEUS code "02-09-09-01"). Note that, consequently, events remedied by means of e.g. retightening, cleaning or refilling are not considered as a failure. In order to compare reliability KPIs of different components, sub-systems and overall turbines, the following average rates are calculated:

corrective maintenance rate 
$$c = \frac{\sum_{i=1}^{I} c_i}{\sum_{i=1}^{I} X_i T_i} = \frac{c}{T}$$
, (1)

preventive maintenance rate 
$$p = \frac{\sum_{i=1}^{I} P_i}{\sum_{i=1}^{I} X_i T_i} = \frac{P}{T}$$
, (2)

failure rate 
$$f = \frac{\sum_{i=1}^{I} N_i}{\sum_{i=1}^{I} X_i T_i} = \frac{N}{T}$$
, (3)

Herein,  $C_i$  is the number of corrective maintenance visits,  $P_i$  is the number of preventive maintenance visits, and  $N_i$  is the number of failures of the analysed component or subsystem in the time interval i.  $X_i$  is the number of WTs analysed within this time interval of duration  $T_i$ . Consequently, the average rates are equal to the quotient of the sum of all corrective, preventive or failure events, C, P and N, respectively, and the total amount of considered WT operational years T.

As WTs of different power classes are included in the analyses, next to average rates per WT and year, average rates per rated capacity in MW and year are also calculated.

Moreover, confidence intervals for the failure rates are computed to quantify the uncertainty stemming from the size of the data sets. According to (Bain and Engelhardt, 1991), the confidence intervals for failure rates based on time-censored data are estimated using Eq. (4):

$$\left[\frac{\chi^{2}(\frac{\alpha}{2}2N)}{2T}, \frac{\chi^{2}(1-\frac{\alpha}{2},2N+2)}{2T}\right] \tag{4}$$

Herein,  $\chi^2(\alpha/2,2N)$  is the  $(\alpha/2)$ -quantile of the  $\chi^2$  distribution with 2N degrees of freedom. In this study,  $\alpha = 0.1$  is utilised to provide confidence intervals with a confidence level of 90%. As explained in more detail in (Fischer et al., 2019a), these confidence intervals based on sample data are to be interpreted in terms of frequency: if a large number of samples (in this

case failure or maintenance data sets covering a part of a WT population) was evaluated, the confidence intervals determined according to Eq. (4) would cover the true value of the failure rate in 90% of the cases.

#### 3.2 Evaluated data sets

The data sets underlying this analysis are based on maintenance reports of onshore and offshore wind turbines. In total, more than 4200 operational years are covered. A detailed overview of the data sets is provided in Table 3. While the offshore data stem from turbines of four different OEMs with turbine capacities ranging up to 9 MW, the onshore data comprise turbines of nine different manufacturers. In total, 1089 WTs located in seven different European countries are considered in the present study.

Table 3: Information about the data sets which have been considered in the analysis.

|                                 | Offshore | Onshore |
|---------------------------------|----------|---------|
| WT operational years considered | 1755     | 2489    |
| Number of WT OEMs covered       | 4        | 9       |
| Rated capacity considered       | Up to    | 9 MW    |
| Available data period           | 2006     | -2024   |

The data set analysed in this study encompasses the following technical concepts:

- Pitch system: hydraulic, electrical
- Drive train concepts: geared, direct drive, hybrid drive
- Generator types: doubly-fed induction generator (DFIG), electrically excited synchronous generator (EESG), permanent magnet synchronous generator (PMSG), squirrel-cage induction generator (SCIG); including low voltage (LV) and medium voltage (MV) generators
- Converter technology: air-cooled, liquid-cooled; including LV and MV converters
- While it is important to include data of both, WTs which have been operated already for some time to analyse failure behaviour over time and WTs which have just recently been commissioned to incorporate newest technologies, this leads to a diverse data set of different turbine generations. The data period analysed in this study is nearly identical for both onshore and offshore WTs. Note that 12.5% of the WTs have a capacity smaller than 2 MW. Most WTs covered within this study can be considered as recent turbine technology.

#### 255 4 Results and discussion

## 4.1 Comparison of failure rates for onshore and offshore wind turbines

Figure 1 and Figure 2 illustrate a comparative analysis of failure rates for onshore and offshore WTs, calculated per WT and year, as well as per MW of turbine capacity and year, respectively. In addition to presenting the average failure rate of the entire WT, Table 5 provides the average failure rates for all 29 subsystems defined by RDS-PP, along with a corresponding translation of RDS-PP codes. For better clarity in the presentation of results, the analysis in this section is limited to the eleven most critical subsystems, selected based on failure frequency. Components that could not be unequivocally assigned to a specific subsystem are categorised under "G", representing "other components". It is important to note that the sum of all subsystem failure rates exceeds the overall WT failure rate, as certain failure events involve the replacement of components across multiple subsystems.

Figure 1: Failure-rate comparison per WT and year of onshore and offshore WTs including the eleven most critical subsystems.

Figure 2: Failure-rate comparison per MW of turbine capacity and year for onshore and offshore WTs including the eleven most critical subsystems.

The comparison of average failure rates per WT and year indicates a higher reliability of onshore WTs (3.3 vs. 4.3 failures per offshore WT and year), consistent with findings frequently reported in the literature (Cevasco et al., 2021). However, when normalised per MW and year, the data reveal that onshore WTs exhibit a higher failure frequency per WT capacity, with an average failure rate of 1.729 failures per MW per year, compared to 1.088 failures per MW per year for offshore WTs. Given the strong dependence of average failure rates on WT size – shown e.g. in (Spinato et al., 2009; Koukoura, 2019; Walgern et al., 2023; Anderson et al., 2025), and also found in our analyses – further analysis and interpretation are based exclusively on failure rates normalised per MW and year. While for onshore WTs the subsystems rotor system (MDA) including the pitch system, the control system (MDY), the drive train system (MDK), and the converter system (MSE) are identified as most critical, for offshore WTs the highest failure rates are recognised for the subsystems rotor system, control system, lifting gears (XMM), and converter system. In previous publications by Fraunhofer IWES, which focused exclusively on the power converter, the converter subsystem also encompassed failures related to main circuit breakers and contactors (cf. (Fischer et al., 2019a; Fischer et al., 2019b; Fischer et al., 2023; Anderson et al., 2025)). In contrast, this study categorises these failures separately within the "Generator Switching System" (MSC) subsystem in order to follow the RDS-PP classification. Additionally, while some of our earlier studies normalised failure rates based on the rated power of the converter, it is important to note that in the present analysis all failure rates, including that of the converter system, are normalised by the rated power of the turbine.

Note that the drive train system covers the subassemblies rotor bearing, speed conversion, drive train brake, high speed shaft, drive train auxiliary systems, main and offline gear oil systems, oil lubrication system, rotor lock, rotor slewing unit, and drive train cooling system. Therefore, the subsystem is evaluated across both WTs with and without gearboxes. A more detailed examination of the MDA system category reveals that for onshore WTs the pitch system accounts for approximately 80.8% of MDA system failures, whereas for offshore WTs it constitutes nearly 82.5% of failures within this category (cf. Table 5). Provided KPIs in Table 5 can be utilised for estimating failures and maintenance interventions. However, it is important to note that the failure behaviour is not solely characterised by turbine size making more sophisticated reliability models necessary to support such analysis.

# 4.2 Failure-rate comparison across WT OEMs

Although it is common practice to report average failure rates derived from mixed fleets comprising different WT types, as presented in Section 4.1, this approach carries inherent risks. Reporting only a group-averaged failure rate without further differentiation might obscure major reliability differences, which can serve as key indicators for root-cause analysis and design optimisation. To address these limitations, an OEM-specific analysis is performed. Figure 3 and Figure 4 present the average failure rates of offshore WTs from four different OEMs and onshore WTs from six different OEMs. Where a manufacturer is included in both Fig. 3 and Fig. 4, they do not share the same label for confidentiality reasons. This means that OEM1 in Fig. 3 is not the same manufacturer as OEM1 in Fig. 4.

Figure 3: Failure-rate comparison per MW and year across WT OEMs of offshore assets.

320

325

Figure 4: Failure-rate comparison per MW and year across WT OEMs of onshore assets.

To ensure that the comparison reflects only technological differences, failure rates are again normalised per MW and year. Analysis results reveal significant disparities in failure rates between WTs from different manufacturers. For offshore WTs, the average failure rate for OEM1 is 1.6 to 2.4 times higher than that of the other three OEMs, with a distinct failure rate of 1.7 failures per MW per year. In the case of onshore WTs, failure rates range between 1.5 and 2.5 failures per MW per year. The variability in confidence intervals reflects the uncertainty associated with the sizes of the underlying data subsets. While datasets for all offshore OEMs and onshore OEMs 2 and 4 include at least 1100 MW-years, analysis for onshore OEMs 1, 3, 5, and 6 are based on smaller datasets ranging from 200 to 330 MW-years. Onshore OEMs 7, 8, and 9 are excluded from this analysis due to insufficient sample sizes. Overall, onshore OEM failure rates generally exceed those of offshore OEMs, with the exception of offshore OEM1, which exhibits a failure rate comparable to the three best-performing onshore OEMs.

# 4.3 Failure-rate behaviour through time

An essential aspect of reliability analysis is the evolution of failure behaviour over time. This is assessed by calculating failure rates across different operating years. To isolate the effect of WT aging, the analysis is conducted for specific WT types, avoiding the confounding influence of mixed turbine designs. As an example, Fig. 5 presents a comparison of normalised failure rates across different operating years, grouped into five periods of WT operating age, for a single WT type including eight representative subsystems.

Figure 5: Comparison of normalised failure rates across different operating years for a specific WT type including eight exemplary subsystems.

The failure rate trajectory for the entire WT system follows the characteristic shape of a bathtub curve (Pulcini, 2001): During the initial years of operation, elevated failure rates are observed, corresponding to early failures. Over time, failure rates decline, reaching a lower and more stable level through operating years 5 to 8. In contrast to the typical shape of the bathtub curve implying low and constant failure rates for a long period of the operational life, this phase is found to be surprisingly short in the investigated WT fleet. From year 9 onward, failure rates increase again, indicative of degradation-related failures. Although confidence intervals show a slight overlap between some groups, the overall trend is clearly visible and observable across different WT types, both onshore and offshore.

The failure behaviour of individual subsystems varies significantly depending on the specific subsystem under analysis. While certain subsystems, such as the drive train system (MDK), yaw system (MDL), and converter system (MSE), exhibit a failure trend similar to that of the overall WT system, others, such as the central hydraulic system (MDX) and the power generation

system (MKA), show a steadily increasing trend suggesting that these are primarily suffering from degradation-related failures. Additionally, some subsystems do not display a distinct trend due to overlapping confidence intervals, either because no distinct trend exists, or the dataset is too limited to detect one. These findings emphasise that the well-established bathtub curve in reliability modelling results from the superposition of different failure mechanisms and trends.

## 4.4 Other O&M relevant KPIs

When utilising reliability data for O&M simulations or OPEX calculations, additional O&M KPIs beyond failure rates are required as input. To address this, further analyses based on the offshore data subset are presented in the following. These include a comparison of corrective and preventive maintenance interventions, an analysis of major component replacements (MCR), and an evaluation of average repair times and the average number of maintenance technicians required per failure event and subsystem. Due to limited access to cost data and the impact of inflation, cost figures for spare parts are not provided, as comparisons across different datasets and years would be challenging. Reference values can be found in (Carroll et al., 2016; BVG Associates, 2019; Stehly et al., 2024).

### 4.4.1 Comparison of corrective and preventive maintenance interventions

Within this study the failure definition is based on the consumption of spare parts, while other corrective maintenance activities not requiring spare parts are classified under the category "Corrective Maintenance other". In addition to addressing failure events and conducting troubleshooting and repairs – both classified as corrective maintenance interventions – technicians are also responsible for preventive maintenance interventions, such as scheduled maintenance. Furthermore, statutory inspections, functional tests, condition monitoring related activities – such as oil sampling – and routine tasks like topping up coolants or lubricants are categorised as preventive maintenance interventions. Figure 6 displays the corresponding maintenance rates per MW and year.

Figure 6: Comparison of corrective and preventive maintenance interventions for offshore wind assets differentiating corrective interventions into failures and other corrective maintenance.

As detailed in Section 5.3, offshore WTs experience an average of 1.088 failures per MW per year. For example, this is equivalent to 5.4 failures per year for a 5 MW turbine and 10.9 failures per year for a 10 MW turbine. Additionally, the category "Corrective Maintenance other" accounts for 1.651 interventions per MW and year, while preventive maintenance actions total 2.664 interventions per MW and year. In total, this results in 5.403 maintenance interventions per MW per year. This translates to approximately 27 maintenance interventions annually for a 5 MW WT and to around 54 maintenance interventions for 10 MW WT. Similar intervention frequencies are observed across offshore wind farms with different WT power classes included in the data sets used for this analysis.

# 4.4.2 Major component replacements

The average failure rates per subsystem presented above are based on all corrective maintenance interventions involving the use of spare parts, regardless of the size or cost of the replaced component. To provide further details relevant for O&M simulations, a distinction is made between total major component replacements (MCR) and those that specifically require a jack-up vessel (JUV), as outlined in (The Crown Estate, 2014). MCR encompasses replacements across six subsystems: the rotor system (MDA), the drive train system (MDK), the power generation system (MKA), the generator transformer system (MST), the nacelle (MUD), and the tower system (UMD). The components considered for each subsystem are listed in Table 4. Average offshore MCR rates as well as rates of interventions requiring a JUV are presented in Table 5.

Table 4: Considered components for major component replacements (MCR) and MCR requiring a jack-up vessel (JUV).

| Subsystem                          | MCR requiring no JUV             | MCR requiring a JUV                         |
|------------------------------------|----------------------------------|---------------------------------------------|
| Rotor system (MDA)                 | -                                | blade, hub, blade bearing                   |
| Drive train system (MDK)           | damaged high and low speed shaft | main bearing, gearbox, rotor shaft assembly |
| Power generation system (MKA)      | generator bearings               | generator                                   |
| Generator transformer system (MST) | -                                | transformer                                 |
| Nacelle (MUD)                      | -                                | nacelle                                     |
| Tower system (UMD)                 | -                                | tower, transition piece, foundation         |

With an average of 0.0209 MCR per MW and year, the power generation system MKA accounts for the highest MCR rate, followed by the drive train system MDK at 0.0149 MCR per MW and year. Of these, 0.0097 MCR per MW and year require a JUV, making the drive train system the primary contributor to MCR events necessitating a JUV. For the rotor system MDA only blade and blade bearing replacements were observed, while no MCR events were recorded for the nacelle, tower, transition piece, or foundation. Across the entire WT, the total MCR rate is 0.0366 per MW and year, with 0.0117 MCR per

https://doi.org/10.5194/wes-2025-212
Preprint. Discussion started: 24 October 2025

MW per year requiring a JUV. For a wind farm comprising 50 WTs, each with a rated capacity of 10 MW, this corresponds to approximately 37% of WTs undergoing a MCR annually, with 12% requiring a JUV – equivalent to roughly six WTs.

### 4.4.3 Average repair time

The average repair time per subsystem is displayed in Table 5. It represents the total duration from the technicians' arrival to their departure from the turbine, regardless of the number of personnel involved in the maintenance intervention. Unlike downtime or time to repair, it does not account for travel time, lead time of spare parts, delays due to inaccessibility, or other external factors (Carroll et al., 2016). It is important to note that the average repair time is calculated across all failure events without distinguishing between failure severity. On average, component replacements for the overall WT system require 2.7 hours. Other corrective maintenance activities take approximately 1.5 hours, while preventive maintenance tasks involve an average technician presence of 3.8 hours.

The longest repair times are observed for the drive train system, rotor system, generator transformer system, and converter system. While extended repair durations are expected for subsystems containing major components, their overall impact on turbine availability remains limited due to relatively low failure rates in most cases. In contrast, the power converter system has a substantial effect on availability, as it exhibits both a high failure rate and prolonged average repair time.

## 4.4.4 Average number of technicians required

Similarly to the average repair time, the average number of technicians required per maintenance intervention for each subsystem is shown in Table 5. This value represents the mean number of technicians who recorded working hours on the WT or were listed in maintenance records. However, this information was available for only half of the WTs in the offshore data set, resulting in a reduced sample size for analysis. Consequently, the data set is insufficient to provide specific figures for MCR beyond the overall averages for all failure events. As a result, the variation in technician requirements across subsystems is relatively small, ranging from 1.9 technicians for the common cooling system to 3.5 technicians for the generator transformer system and generator switching system. On average, 2.5 technicians are required for both other corrective maintenance activities and preventive maintenance interventions.

# 415 4.5 Comparison with results from literature

Although a direct comparison with existing literature is not feasible due to variations in turbine sizes, technologies, and generations considered in different studies, this section aims to contextualise the findings of this paper within the existing body of reliability and O&M research. For offshore WTs, studies by the University of Strathclyde (Carroll et al., 2016) and SPARTA (SPARTA, 2017; SPARTA, 2022) are referenced, while for onshore turbines, comparisons are drawn with findings from WMEP (Faulstich et al., 2011), Reliawind (Gayo, 2011) and AWESOME (Reder et al., 2016). However, direct comparisons remain challenging due to differences in categorisation systems and variations in KPI definitions. For example, Carroll et al. report annual failure rates, whereas SPARTA provides monthly repair rates. This shows that the definition of failure itself

varies across studies. (Anderson et al., 2023) emphasise that such differences in failure definitions in field-data-based studies significantly impact the reported KPI values. Despite these challenges, a general comparison remains valuable to place our results in the context of other research work.

For onshore WTs, an overall average failure rate of 1.729 failures per MW and year has been determined in the present study, with the pitch system, control system, drive train system, and converter system identified as the most critical subsystems. Similar findings were reported by (Faulstich et al., 2011), who calculated an annual failure rate of 2.4 failures per WT – consistent with the smaller rated capacities of the turbines in their study – while also highlighting the electrical and control systems as particularly critical. Although (Gayo, 2011) reported only normalised failure rates, their findings similarly identified the power module (including power converter, generator, transformer and switchgears), rotor module (including pitch system, blades and hub), control system, and drive train system among the five most frequently failing subsystems. In contrast, (Reder et al., 2016) highlight the gearbox, the blades, the blade brake, generator, and controller as most critical, while reporting lower normalised failure rates for the pitch system and the frequency converter.

For offshore WTs, an annual average failure rate of 1.088 per MW have been determined in this study. (Carroll et al., 2016) reported approximately 7.8 failures per turbine per year, including major component replacements, as well as major and minor repairs, for turbines with rated capacities between 2 and 4 MW. Transforming the findings of our study to a 3 MW turbine results in an estimated 3.3 failures per turbine and year, which appears significantly lower. However, considering discrepancies in failure definitions and incorporating the additional 1.651 interventions per MW per year associated with corrective maintenance interventions beyond component failures, the estimated corrective maintenance rate reaches approximately 8.2 for a 3 MW turbine – closely aligning with the figures reported by (Carroll et al., 2016). This highlights the substantial impact that failure definitions and the inclusion criteria for corrective maintenance activities have on reported failure rates.

Regarding the most failure-prone subsystems of offshore WTs, the pitch system, control system, and converter system have emerged as critical in our study, consistent with the top four failing subsystems identified in the (SPARTA, 2017) report. Similarly, (Carroll et al., 2016) highlighted the pitch system as a major contributor to failure events. Furthermore, significant differences in annual failure rates were observed across different OEMs, a finding also noted by (SPARTA, 2022) when comparing forced outages per turbine between two OEMs for selected subsystems.

The analysis has also revealed variations in failure behaviour over time, with the overall WT system following the characteristic bathtub curve. At the same time, different subsystems exhibit different failure trends. (Faulstich et al., 2011) reported a similar trend for overall onshore WTs. (SPARTA, 2022) assessed temporal patterns for repairs for specific components and subsystems not directly comparable with failure events and their trends evaluated within this study. The increasing repair rate observed for the generator in the SPARTA evaluation aligns with the trends found for the power generation system (MKA) in the present study, whereas other subsystems are not directly comparable due to differences in component classification.

Regarding major component replacements, both this study and (Carroll et al., 2016) identified the power generation system and drive train system as the primary contributors to JUV interventions. Finally, reported average repair times and the number of technicians required for replacements were compared with findings from (Carroll et al., 2016). Repair times in this study

were generally lower than those reported by Carroll et al., even when compared with Carroll's "minor repairs" category, which primarily includes small spare parts driving overall failure rates. On average, 2.8 technicians were required per replacement according to our results, which is in the same range as the figures reported in (Carroll et al., 2016).

|                                      |                    |                                            |                                                         |                                              |                                             |                                                         |                                        |                                         |                                                |                                              |                           | 2                            |                             |
|--------------------------------------|--------------------|--------------------------------------------|---------------------------------------------------------|----------------------------------------------|---------------------------------------------|---------------------------------------------------------|----------------------------------------|-----------------------------------------|------------------------------------------------|----------------------------------------------|---------------------------|------------------------------|-----------------------------|
| Subsystem                            | RDS-<br>PP<br>Code | Average<br>Failure<br>Rate<br>On-<br>shore | Lower<br>bound<br>of 90%<br>confi-<br>dence<br>interval | Upper bound of 90% confidence dence interval | Average<br>Failure<br>Rate<br>Off-<br>shore | Lower<br>bound<br>of 90%<br>confi-<br>dence<br>interval | Upper bound of 90% confidence interval | Average<br>Rate<br>MCR<br>Off-<br>shore | Average<br>Rate<br>MCR<br>JUV<br>req-<br>uired | Average<br>tech-<br>nicians<br>req-<br>uired | Average<br>repair<br>time | tive Maintenance Rate (excl. | Preventive Maintenance Rate |
| Unit                                 |                    | 1/<br>(MW·a)                               | 1/<br>(MW·a)                                            | 1/<br>(MW·a)                                 | 1/<br>(MW·a)                                | 1/<br>(MW·a)                                            | 1/<br>(MW·a)                           | 1/<br>(MW·a)                            | 1/<br>(MW·a)                                   | 1/<br>WT visit                               | h/<br>WT visit            | 1/<br>(MW·a)                 | 1/<br>(MW·a)                |
| Onshore / offshore                   | ffshore            | onshore                                    | onshore                                                 | onshore                                      | offshore                                    | offshore                                                | offshore                               | offshore                                | offshore                                       | offshore                                     | offshore                  | offshore                     | offshore                    |
| Environmental<br>Measuring System    | I CKJ              | 0.0590                                     | 0.0534                                                  | 0.0652                                       | 0.0272                                      | 0.0240                                                  | 0.0307                                 |                                         |                                                | 2.8                                          | 2.6                       |                              |                             |
| Rotor Rotor<br>System System         |                    | 0.0773                                     | 0.0708                                                  | 0.0843                                       | 0.0379                                      | 0.0341                                                  | 0.0420                                 | 900000                                  | 0.0006                                         | 3.2                                          | 8.                        |                              |                             |
| mcl.<br>Pitch Pitch<br>System System | MDA<br>sh<br>em    | 0.3138                                     | 0.3006                                                  | 0.3275                                       | 0.1638                                      | 0.1559                                                  | 0.1721                                 | ı                                       | 1                                              | 3.2                                          | 3.3                       | ı                            | ı                           |
| Drive Train System                   | em MDK             | 0.2281                                     | 0.2168                                                  | 0.2398                                       | 0.0921                                      | 0.0861                                                  | 0.0983                                 | 0.0149                                  | 0.0097                                         | 2.9                                          | 9.7                       | 1                            | ı                           |
| Yaw System                           | MDL                | 0.0846                                     | 0.0778                                                  | 0.0919                                       | 0.0784                                      | 0.0730                                                  | 0.0842                                 | ı                                       | ı                                              | 3.0                                          | 2.9                       |                              | 1                           |
| Central Hydraulic<br>System          | lic MDX            | 0.0599                                     | 0.0542                                                  | 0.0660                                       | 0.0806                                      | 0.0751                                                  | 0.0865                                 | ı                                       | 1                                              | 3.1                                          | 2.7                       | 1                            | ı                           |
| Control System                       | n MDY              | 0.2548                                     | 0.2429                                                  | 0.2671                                       | 0.1268                                      | 0.1198                                                  | 0.1341                                 | 1                                       | 1                                              | 3.2                                          | 2.7                       | 1                            | 1                           |
| Power Generation<br>System           | on MKA             | 0.1206                                     | 0.1124                                                  | 0.1292                                       | 0.0835                                      | 0.0779                                                  | 0.0895                                 | 0.0209                                  | 0.0012                                         | 2.5                                          | 2.9                       | •                            | •                           |
| Generator<br>Switching System        | m MSC              | 0.0670                                     | 0.0610                                                  | 0.0735                                       | 0.0250                                      | 0.0219                                                  | 0.0283                                 | ı                                       | ı                                              | 3.5                                          | 2.1                       | ı                            | ı                           |
| Converter System                     | m MSE              | 0.2226                                     | 0.2115                                                  | 0.2342                                       | 0.1243                                      | 0.1174                                                  | 0.1315                                 | 1                                       | 1                                              | 3.4                                          | 4.2                       | ı                            | ı                           |
| Generator<br>Transformer<br>System   | MST                | 0.0252                                     | 0.0215                                                  | 0.0293                                       | 0.0083                                      | 9900.0                                                  | 0.0103                                 | 0.0003                                  | 0.0003                                         | 3.5                                          | 4.3                       |                              |                             |
|                                      |                    |                                            |                                                         |                                              |                                             |                                                         |                                        |                                         |                                                |                                              |                           |                              |                             |

|                                                 |             |                                            | I own                              | Linnon                           |                                             | I course                         | Linnor                                                  |                                         | Ayorogo                             |                                              |                           | Correc-                                    |                             |
|-------------------------------------------------|-------------|--------------------------------------------|------------------------------------|----------------------------------|---------------------------------------------|----------------------------------|---------------------------------------------------------|-----------------------------------------|-------------------------------------|----------------------------------------------|---------------------------|--------------------------------------------|-----------------------------|
| Subsystem                                       | RDS-PP Code | Average<br>Failure<br>Rate<br>On-<br>shore | bound of 90% confi- dence interval | bound of 90% confidence interval | Average<br>Failure<br>Rate<br>Off-<br>shore | bound of 90% confidence interval | opper<br>bound<br>of 90%<br>confi-<br>dence<br>interval | Average<br>Rate<br>MCR<br>Off-<br>shore | Rate<br>MCR<br>JUV<br>req-<br>uired | Average<br>tech-<br>nicians<br>req-<br>uired | Average<br>repair<br>time | tive<br>Main-<br>tenance<br>Rate<br>(excl. | Preventive Maintenance Rate |
| Unit                                            |             | 1/<br>(MW·a)                               | 1/<br>(MW·a)                       | 1/<br>(MW·a)                     | 1/<br>(MW·a)                                | 1/<br>(MW·a)                     | 1/<br>(MW·a)                                            | 1/<br>(MW·a)                            | 1/<br>(MW·a)                        | 1/<br>WT visit                               | h/<br>WT visit            | 1/<br>(MW·a)                               | 1/<br>(MW·a)                |
| Onshore / offshore                              | ıre         | onshore                                    | onshore                            | onshore                          | offshore                                    | offshore                         | offshore                                                | offshore                                | offshore                            | offshore                                     | offshore                  | offshore                                   | offshore                    |
| Nacelle                                         | MUD         | 0.0267                                     | 0.0229                             | 0.0309                           | 0.0202                                      | 0.0175                           | 0.0232                                                  | 0.0000                                  | 0.0000                              | 3.3                                          | 1.9                       | 1                                          | ı                           |
| Remote Monitoring<br>System                     | MYA         | 0.0071                                     | 0.0053                             | 0.0095                           | 0.0004                                      | 0.0001                           | 0.0011                                                  | 1                                       | ı                                   | 2.0                                          | 2.2                       | 1                                          | ı                           |
| Tower System                                    | UMD         | 0.0204                                     | 0.0171                             | 0.0241                           | 0.0151                                      | 0.0128                           | 0.0178                                                  | 0.0000                                  | 0.0000                              | 2.7                                          | 2.5                       | 1                                          | ı                           |
| Personnel Rescue<br>Systems                     | WBA         | 0.0059                                     | 0.0042                             | 0.0081                           | 0.0019                                      | 0.0011                           | 0.0030                                                  | •                                       | 1                                   | 2.2                                          | 1.4                       | 1                                          | ı                           |
| Fire Extinguishing<br>System                    | XGM         | 0.0078                                     | 0.0058                             | 0.0102                           | 0.0041                                      | 0.0029                           | 0.0056                                                  | ı                                       | ı                                   | 2.9                                          | 2.3                       | ı                                          | ı                           |
| Lifting Gears                                   | XMM         | 0.0246                                     | 0.0210                             | 0.0287                           | 0.1255                                      | 0.1185                           | 0.1327                                                  | ı                                       | ı                                   | 2.5                                          | 2.8                       | 1                                          | ı                           |
| Obstacle Warning<br>System                      | XSD         | 0.0328                                     | 0.0286                             | 0.0374                           | 0.0222                                      | 0.0194                           | 0.0254                                                  | ı                                       | 1                                   | 2.6                                          | 1.9                       | •                                          | ı                           |
| Low Voltage<br>Electrical Main<br>Supply System | BFA         | 0.0006                                     | 0.0002                             | 0.0016                           | 0.0243                                      | 0.0213                           | 0.0276                                                  | 1                                       | ı                                   | 2.6                                          | 1.2                       | ı                                          | ī                           |
| Fire Alarm System                               | CKA         | 0.0044                                     | 0.0030                             | 0.0064                           | 0.0155                                      | 0.0132                           | 0.0183                                                  | ı                                       | 1                                   | 3.4                                          | 3.0                       | ı                                          | 1                           |
| Transformer<br>Station                          | UAB         | 0.0025                                     | 0.0015                             | 0.0041                           | ı                                           | ı                                | 1                                                       | 1                                       | ı                                   | 1                                            | ,                         | ,                                          | ı                           |
| Equipotential<br>Bonding / Earthing<br>System   | XFB         | 0.0029                                     | 0.0018                             | 0.0046                           | 0.0049                                      | 0.0036                           | 9900.0                                                  | ı                                       | ı                                   | 2.2                                          | 2.6                       | 1                                          | ı                           |
| Lightning<br>Protection System                  | XFC         | 0.0025                                     | 0.0015                             | 0.0041                           | 9600.0                                      | 0.0077                           | 0.0118                                                  | 1                                       | ı                                   | 2.4                                          | 1.3                       |                                            | 1                           |
|                                                 |             |                                            |                                    |                                  |                                             |                                  |                                                         |                                         |                                     |                                              |                           |                                            |                             |

|                                  |                    |                                            | Lower                            | Hanor                                                   |                                             | I course                         | Llnnor                                                  |                                         | Ayorogo                                        |                                              |                           | Correc-                                    |                             |
|----------------------------------|--------------------|--------------------------------------------|----------------------------------|---------------------------------------------------------|---------------------------------------------|----------------------------------|---------------------------------------------------------|-----------------------------------------|------------------------------------------------|----------------------------------------------|---------------------------|--------------------------------------------|-----------------------------|
| Subsystem                        | RDS-<br>PP<br>Code | Average<br>Failure<br>Rate<br>On-<br>shore | bound of 90% confidence interval | opper<br>bound<br>of 90%<br>confi-<br>dence<br>interval | Average<br>Failure<br>Rate<br>Off-<br>shore | bound of 90% confidence interval | opper<br>bound<br>of 90%<br>confi-<br>dence<br>interval | Average<br>Rate<br>MCR<br>Off-<br>shore | Average<br>Rate<br>MCR<br>JUV<br>req-<br>uired | Average<br>tech-<br>nicians<br>req-<br>uired | Average<br>repair<br>time | tive<br>Main-<br>tenance<br>Rate<br>(excl. | Preventive Maintenance Rate |
| Unit                             |                    | 1/<br>(MW·a)                               | 1/<br>(MW·a)                     | 1/<br>(MW·a)                                            | 1/<br>(MW·a)                                | 1/<br>(MW·a)                     | $1/(MW \cdot a)$                                        | $1/(MW \cdot a)$                        | 1/<br>(MW·a)                                   | 1/<br>WT visit                               | h/<br>WT visit            | $1/(MW \cdot a)$                           | 1/<br>(MW·a)                |
| Onshore / offshore               | ore                | onshore                                    | onshore                          | onshore                                                 | offshore                                    | offshore                         | offshore                                                | offshore                                | offshore                                       | offshore                                     | offshore                  | offshore                                   | offshore                    |
| Ventilation<br>Systems           | XAM                | 1                                          |                                  |                                                         | 0.0029                                      | 0.0019                           | 0.0042                                                  |                                         | 1                                              | 2.2                                          | 8.1                       | 1                                          |                             |
| Central<br>Lubrication<br>System | MDV                | 0.0218                                     | 0.0184                           | 0.0257                                                  | 0.0015                                      | 0.0008                           | 0.0025                                                  | 1                                       | 1                                              | 2.5                                          | 1.9                       | 1                                          |                             |
| Compensation<br>System           | MSS                | 0.0084                                     | 0.0063                           | 0.0109                                                  | 0.0004                                      | 0.0001                           | 0.0011                                                  | ı                                       | ı                                              | ı                                            | ı                         | ı                                          |                             |
| Cooling System                   | MUR                | 0.0078                                     | 0.0058                           | 0.0102                                                  | 0.0026                                      | 0.0017                           | 0.0039                                                  | ı                                       | ı                                              | 1.9                                          | 2.2                       | ı                                          |                             |
| Telephone System                 | *                  | 0.0204                                     | 0.0171                           | 0.0241                                                  | 0.0064                                      | 0.0049                           | 0.0082                                                  | ı                                       | ı                                              | 2.0                                          | 6.0                       | ı                                          |                             |
| General / Other                  | ß                  | 0.1210                                     | 0.1128                           | 0.1296                                                  | 0.0579                                      | 0.0533                           | 0.0629                                                  | 1                                       | ı                                              | 3.2                                          | 2.6                       | ı                                          | 1                           |
| Wind Turbine<br>(WT) overall     | WT                 | 1.7286                                     | 1.6973                           | 1.7602                                                  | 1.0875                                      | 1.0669                           | 1.1084                                                  | 0.0366                                  | 0.0117                                         | 2.8                                          | 2.7                       | 1.6508                                     | 2.6637                      |

#### 5 Conclusions and outlook

This study provides a comprehensive analysis of failure rates for offshore and onshore wind turbines (WTs), as well as repair times and maintenance resource requirements for offshore assets, with a particular focus on subsystem-level reliability. Based on real-world maintenance data from over 1000 onshore and offshore WTs covering more than 4200 operational years, this data set offers unique diversity, size and recentness when compared to those used in previous reliability studies. The results highlight that while onshore WTs exhibit lower failure rates per turbine and year, their failure rates per megawatt and year are 470 higher compared to offshore WTs. Given the strong dependence of failure rates on the turbines' rated power, further analyses have been conducted based on failure rates per MW and year to ensure comparability. Onshore WTs exhibit an average failure rate of 1.729 failures per MW per year, whereas offshore WTs demonstrate a lower annual average failure rate of 1.088 failures per MW.

The analysis of subsystem-level failure rates has revealed that certain components, such as the pitch system (0.314 vs. 0.164 475 failures per MW and year), the control system (0.255 vs. 0.127 failures per MW and year), and the converter system (0.223 vs. 0.124 failures per MW and year), contribute disproportionately to overall WT unreliability for both onshore and offshore turbines. While the drive train system exhibited notably high failure rates for onshore WTs, offshore WTs experienced elevated failure rates in the lifting gear system. Particularly the power converter system has been identified as a critical subsystem due to its combination of a high average failure rate and extended repair duration, making it a major factor affecting overall WT 480 availability next to long-lasting replacement campaigns of major components. Additionally, major component replacements (MCR) have been analysed, distinguishing between those requiring a jack-up vessel (JUV) and those that do not. The power generation system and drive train system accounted for the majority of MCRs, with the latter also being responsible for the highest share of JUV-requiring replacements.

The study has also examined failure behaviour through time, demonstrating that the overall WT failure pattern follows the well-established bathtub curve, with high early failure rates, a period of stability, and increasing failure rates due to degradation in later years of turbine operation. However, subsystem-specific trends vary, with some following the same pattern as the overall WT and others dominated by degradation failures or displaying no clear trend.

In addition to failure rates, i.e. the frequency of corrective measures including spare-part consumption, corrective maintenance interventions without spare-part use and preventive maintenance tasks have also been analysed. On average, 2.7 hours are required for component replacements, while other corrective maintenance and preventive maintenance activities take 1.5 hours and 3.8 hours, respectively. The number of technicians required per maintenance intervention varies by subsystem, ranging from 1.8 to 3.5 technicians, with an overall average of 2.5 technicians per other corrective and preventive maintenance task. While major component failures have significant repair times, their relatively low failure rates limit their impact on availability. In contrast, frequently failing subsystems such as the power converter system have a substantial influence on turbine performance and should be prioritised in reliability-driven design improvements.

https://doi.org/10.5194/wes-2025-212

Preprint. Discussion started: 24 October 2025

© Author(s) 2025. CC BY 4.0 License.

european academy of wind energy

WIND

ENERGY

SCIENCE

DISCUSSIONS

Our findings emphasise the importance of detailed, subsystem-level reliability analyses to enhance the accuracy of O&M simulations and operational expenditure (OPEX) calculations. Aggregated failure rates derived from mixed turbine fleets may obscure critical differences in reliability between turbine types, underscoring the necessity of subgroup-specific analyses. At the same time, the coverage of a variety of WT types and manufacturers is an important prerequisite for providing representative results.

Ultimately, this study underscores the complexity of WT reliability and maintenance planning, highlighting the need for continued field-data based analysis to optimise O&M strategies and improve the long-term sustainability of wind energy operations. Future research will extend beyond basic failure rate calculations to develop advanced reliability models that capture temporal trends in failure behaviour and quantify the effect of various factors on reliability, including design aspects and operating conditions.

#### **Author contribution**

Conceptualisation: J.W., K.F.; Data curation: J.W., N.S., M.H., N.T., M.M., F.A., K.F.; Formal analysis: J.W.; Funding acquisition: J.W., K.F., A.K.; Investigation: J.W.; Methodology: J.W.; Project administration: J.W.; Supervision: K.F., A.K., J.W.; Validation: J.W., K.F.; Visualisation: J.W.; Writing (original draft preparation): J.W., A.K.; Writing (review and editing): K.F., A.K., F.A., J.W.

## **Competing interests**

At least one of the (co-)authors is a member of the editorial board of Wind Energy Science.

# Acknowledgements

The provision of comprehensive field data by project partners is gratefully acknowledged.

# 515 Financial support

The present work was mostly carried out within the research project "Reduction of uncertainties for continued operation of offshore wind farms combining reliability and yield analysis (RUN25+)" funded by the German Federal Ministry for Economic Affairs and Climate Action (BMWK) under grant number 03EE3106. Further financial support was received by EPSRC through the Wind and Marine Energy Systems Centre for Doctoral Training under the grant number EP/S023801/1.

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
