# Peer review of "Reliability and O&M key performance indicators of onshore and offshore wind turbines based on field-data analysis"

_Wind Energy Science, 2025_

## Referee Comment (RC1)

This manuscript presents a comprehensive analysis of reliability and Operations & Maintenance (O&M) data from a significant fleet of over 1000 onshore and offshore wind turbines, covering more than 4200 operational years. The study addresses a persistent gap in wind energy literature, which is the lack of recent, high-quality, and standardized field data for modern multi-megawatt turbines.

The paper is well-structured, the methodology is robust, and the results regarding failure rates, repair times, and technician requirements provide high value to the research and industrial community. The distinction between "failures" (requiring spare parts) and "other corrective maintenance" is a crucial differentiation that adds depth to the analysis.

However, the choice to present the primary comparative analysis normalized by "per MW" rather than the traditional "per turbine" warrants deeper discussion. While the authors justify this based on the dependence of failure rates on turbine size, this linear normalization may obscure logistical realities for O&M planning. Below, my comments and suggestions follow:

**1. Normalization of Failure Rates (Per Turbine vs. Per MW):**

The authors state that onshore wind turbines have higher failure rates per megawatt than offshore wind turbines, while they have lower failure rates per turbine. Even though this is an interesting observation, the decision to focus the remainder of the analysis and conclusions on the "per MW" metric is significant.

From an O&M logistics perspective (vessel transfers, crew planning), a "visit" is a discrete event regardless of the turbine's capacity. A 10 MW turbine failing does not necessarily require twice the number of visits as a 5 MW turbine, though the cost of the spare part or the revenue loss might scale. However, throughout the manuscript, the authors either implicitly or explicitly assume a linear scaling of failure rates versus turbine size. As an example, in line 368: "As detailed in Section 5.3, offshore WTs experience an average of 1.088 failures per MW per year. For example, this is equivalent to 5.4 failures per year for a 5 MW turbine and 10.9 failures per year for a 10 MW turbine."

The authors should elaborate on the justification for assuming a linear scaling of failure rates with capacity. Does a control system or a sensor really fail more frequently just because it is installed on a larger machine? It is recommended to include a brief discussion on which subsystems scale linearly with power (e.g., perhaps converters) and which do not, to prevent readers from misinterpreting the "per MW" metric as a universal rule for reliability scaling.

Given the large amount of data available, a quantitative analysis of this trend seems important. This analysis must be performed separately for onshore and offshore turbines to isolate the effect of operation environment and focus on the effect of size.

**2. The "Short" Useful Life Phase:**

The analysis of failure behaviour over time identifies a bathtub curve. However, the authors note that the constant failure rate phase ("useful life") is "surprisingly short," lasting only from years 5 to 8 before degradation sets in. This is a critical finding for life extension and financial modelling.

Could the authors provide more hypothesis or insight into why the degradation phase begins so early (Year 9)? Is this driven by specific components (e.g., pitch hydraulics or blade erosion) that have shorter lifecycles than the 20-year design life? Highlighting the specific subsystems driving this early rise in the wear-out phase (as hinted at in Figure 5) in the text would strengthen the practical implications of the paper.

**3. Minor comments:**

1. Figures 3 and 4 use colour coding for OEMs. Ensure these colours are distinguishable in black-and-white prints or for colour-blind readers.
2. Line 276 "Given the strong dependence of average failure rates on WT size": This sentence is a bit ambiguous. It is better to explicitly state what kind of dependence is meant (larger turbines fail more often).

**Final remarks:**

This is a high-quality manuscript that provides much-needed transparency on wind turbine reliability using a large, recent dataset. The move toward standardizing data processing and differentiating between spare-part consumption and general maintenance is commendable. With the clarifications requested, this paper can become a benchmark reference for the industry.